# Safety and Pharmaceutical Evaluation of a Novel Natural Polymer, Ocicum, as Solubility and Dissolution Enhancer in Solid Dispersion

**DOI:** 10.3390/ph15070869

**Published:** 2022-07-14

**Authors:** Mobina Manzoor, Syed Atif Raza, Mulazim Hussain Asim, Nadeem Irfan Bukhari, Shumaila Arshad, Uzma Zafar

**Affiliations:** 1Punjab University College of Pharmacy, University of the Punjab, Lahore 54000, Pakistan; mobina_star@hotmail.com (M.M.); uzmazafar261@gmail.com (U.Z.); 2Institute of Pharmacy, Lahore College for Women University, Lahore 54000, Pakistan; 3College of Pharmacy, University of Sargodha, Sargodha 40100, Pakistan; mulazim.hussain@uos.edu.pk; 4Department of Pharmaceutical Technology, Faculty of Pharmacy, Charles University, 50005 Hradec Králové, Czech Republic; arshads@faf.cuni.cz

**Keywords:** aqueous solubility, natural polymer, Ocicum, Poloxamer 407, solid dispersion

## Abstract

Plant mucilages are commonly employed as excipients in pharmaceutical manufacturing. *Ocimum basilicum* (Lamiaceae family), a source of hydrophilic mucilage referred herein as Ocicum, was evaluated for the solubility enhancer of a model drug, aceclofenac, in solid dispersions prepared using different methods. Polymer was extracted from *O. basilicum* and solid dispersions of aceclofenac were fabricated with Ocicum or Poloxamer 407 using polymer-to-drug ratios of 1:1, 1:2 and 1:3 utilizing solvent evaporation, lyophilization and melt methods. Ocicum was evaluated for its safety via acute toxicity study including different biochemical and hematological parameters including liver and kidney profiles. Moreover, different characterization studies including melting-point, Fourier transform infrared spectroscopy (FTIR), X-ray diffraction (XRD), scanning electron microscopy (SEM), differential scanning calorimetry (DSC) and differential thermal analysis (TGA) were used for evaluation of polymer and solid dispersions. Furthermore, solubility and dissolution studies were performed to confirm solubility enhancement. Ocicum was found to be safer, and different characterization studies confirmed the purity of the compounds. In addition, Ocicum exhibited up to 6.27-fold enhanced solubility as compared to pure aceclofenac; similarly, 4.51-fold increased solubility by the synthetic polymer in their respective solid dispersions was shown. Furthermore, Ocicum-based solid dispersions showed substantial improvement in dissolution of aceclofenac. Therefore, it can be concluded from the above-mentioned results that Ocicum might be used as an economical natural oral delivery carrier alternative to the synthetic polymers.

## 1. Introduction

New and better excipients are continuously being introduced to meet the demands of conventional and advanced drug-delivery systems [1]. Natural products, materials obtained from plants, vertebrates and invertebrates are preferred as excipients in pharmaceutical manufacturing because of their easy and ample availability, compatibility, biodegradability, safety and possibility of functional modifications relative to synthetic excipients. Plant products such as mucilages and gum exudates are excellent substitutes of synthetic substances as excipients in pharmaceutical dosage forms due to the above reasons, as well as their low cost and being environment friendly [2,3]. Basil seeds (*Ocimum basilicum*), an annual herb belonging to the Lamiaceae family, is indigenous to Pakistan, India and Iran and is acclimatized to the tropical, subtropical and temperate regions of the world. It grows on the lower hills of Himalaya and is widely cultivated in the tropical regions [4]. *Ocimum* seeds are commonly used for the treatment of dyspepsia, colic ulcers, diarrhea and in traditional desserts, while the herb is famous for its aromatic constituents and used for the culinary purposes. A coarse powder from *Ocimum basilicum* seeds has been analyzed as the pharmaceutical excipient for its physiochemical properties, water-holding capacity and phytochemical characteristics [3]. Sufficient hydration capacity, pertinent to mucilage adhered on seeds, has been its remarkable feature. In early studies, cold-water-extracted polysaccharides from Basil gum were shown to be composed of glucomannan (43%) having a glucose-to-mannose ratio of 10:2 and 1–4 linked xylan (24.29%) with a minor fraction of glucan (2.3%). According to the Committee on the Safety of Pharmaceutical Excipients, a pharmaceutical excipient must be safe to qualify this polymer as a suitable pharmaceutical excipient. Nevertheless, the safety and toxicity of the natural polymer under study has not yet been studied. Furthermore, the mucilages/hydrogels are hydrocolloids; they may be used as the carrier to enhance the solubility and dissolution of poorly soluble drugs belonging to BCS class II. Thus, an evaluation of the polymer from *Ocimum basilicum* seeds as a solubility and dissolution enhancer could be of worth. Nevertheless, like the potential toxicity of the polymer, its use for solubility enhancement has not been studied.

The drug solubility facilitates transforming solid-dosage forms in solution state in aqueous gastrointestinal fluid, leading to convenience of entrance of the drug in blood [5]. Thus, solubility of the drugs in the gastrointestinal tract is a prerequisite for systemic absorption of oral-dosage forms. Almost 70% of the total drugs belong to BCS class II drugs; therefore, they have to be administered in large quantities to achieve therapeutic blood levels. Higher doses not only increase the chances of toxicity but also unnecessarily enhance the burden of cost of therapy [6]. A number of the following techniques are employed to improve the solubility of drugs: microencapsulation, particle-size reduction, co-crystallization [7] and surfactant-assisted wet granulation [8]. Additionally, nanoparticle formation [9], liposomes and niosomes and dendrimers can also be used for amelioration of drug solubility. 

Poor solubility problems can also be handled by formulating solid dispersions with suitable hydrophilic polymers, wherein one or more active ingredients are dispersed in an inert polymeric or other matrix in solid state. Solid dispersion by modifying the drug’s solid-state properties, besides leading to enhanced solubility, also improves dissolution rate and stability. These are among the simplest dosage forms, and may contain the drug and polymer in a binary solid-dispersion system while additionally containing surfactant in a tertiary system. Solid dispersions can be fabricated by using different methods, such as solvent evaporation [6], melting method [10], spray drying [11], hot melt extrusion, supercritical fluid technology [12] and electrostatic spinning [13]. However, one of the most promising and cheap methods is solvent evaporation, which involves digestion of the drug and carrier in single or multiple solvents to make a clear solution, followed by solvent evaporation [5,14]. During evaporation, the crystalline component may unite again to form crystals [15], which could effectively be averted by the polymer present in the solution by preventing nuclei formation with the rising viscosity and keeping the molecular component trapped. The type and amount of carrier are critical for inhibiting recrystallization. In the melting method, the drug and polymer are melted to a liquid state and cooled until solidification. In lyophilization, the drug and polymer are added in solvent and further subjected to freezing to transform into a lyophilized dispersion. Induction of natural polymers in solid dispersion for enhancement of solubility is a less emphasized research area. 

Aceclofenac exhibits poor bioavailability, which results from its practical insolubility, being a BCS class II drug. Improvement in the solubility and dissolution of aceclofenac can play a key role in its bioavailability enhancement. Thus, aceclofenac could be a model drug for evaluation of mucilage of *Ocimum basilicum* as enhancer of solubility and dissolution. Aceclofenac is an anti-inflammatory drug, nonsteroidal in nature and effective in reducing inflammation by nonselective inhibition of cyclooxygenase and specific inhibition of COX-2. It also inhibits prostaglandin (PGE2), cytokines, interleukins and tumor necrosis factor. Hence, in the present study, polymer extracted from *Ocimum basilicum*, named herein as Ocicum, was evaluated for its safety and functionality in formulation design to enhance solubility/dissolution using a solid-dispersion approach with aceclofenac. Moreover, this study focuses on the novel utilization of extracted polymer as a solubility and dissolution enhancer, as well as physical characterization and biological evaluation in terms of toxicity studies. To best of our knowledge, such studies have never been reported, as the use of Ocicum as polymer has never been documented.

## 2. Results and Discussion

In the current research, a natural polymer, Ocicum, from *Ocimum basilicum* was extracted and evaluated for safety, enabling its utility as excipient in drug-delivery systems. To our best knowledge, the Ocicum has been examined for safety profiling and as polymeric excipient for enhancing solubility and dissolution for the first time. This Ocicum was employed in fabrication of the ACF binary solid dispersions with varying ratios by three different methods: the solvent evaporation, lyophilization and melt method, and its performance were compared with synthetic polymer, i.e., Poloxamer 407. The binary solid dispersions are simpler formulations, as they contain drug and polymer to capture the effect only of the polymer on the solubility and the dissolution of aceclofenac. Poloxamer 407, a synthetic polymer was selected for comparison, as it has shown significant improvement in the solubility and dissolution of aceclofenac [16] and other poorly soluble drugs [17,18,19].

### 2.1. Characteristics of Ocicum

The dried mucilage in powder form from the various experimental runs was weighed, and yield was found to be 4 to 7%. The melting point was found to be 122 to 124 °C in different experimental runs. The main reason for the broad melting temperature is that one polymer is composed of different chain lengths that have different times to melt since they require different energies to melt. FTIR spectra of the polymer (Figure 1A) from different experimental runs revealed peaks occurring at 3346 cm^−1^, 1596–1592 cm^−1^, 1402–1400 cm^−1^ and 1033–1032 cm^−1^. A broad absorption band in the range of 3400–3200 cm^−1^ was because of the OH stretch, indicating the hydrophilic nature of the extracted Ocicum; no peak in the range of 2500–2000 cm^−1^ signified the absence of the triple bond, whereas no peak in the range of 1900–1700 cm^−1^ meant the absence of the keto group. As N-H bending of the amide bond occurred at 1650–1580 cm^−1^, the peak at 1592 cm^−1^ might therefore be attributed to the presence of carbonyl groups in protein in the dried mucilage. The OH bending of carboxylic acid occurred at 1440–1395 cm^−1^ so the peak at 1400 cm^−1^ might be because of the fatty acid. Additionally, because of the C=C stretch, the peak appeared between 1650–1566 cm^−1^, and thus might be because of the presence of unsaturation in fatty acid [20]. Similarly, the peak around 3300 cm^−1^ confirms the C-H bond starching of fatty acid. Moreover, spectra of different experimental runs were superimposable to each other, endorsing uniformity among them (Figure 1A). 

Differential scanning calorimetry analysis of Ocicum (Figure 1B) showed an endotherm at 122–124 °C indicating its melting point. A broad endothermic peak hinted on a semi-crystalline nature of the polymer. As the temperature rose from 160–246 °C, the baseline moved upward and the endotherm appeared at 267.5 °C, revealing conformational changes in molecular structure and the release of energy in the form of heat by the polymer. It might be attributed towards chemical reaction or recrystallization at elevated temperature. TGA analysis of Ocicum given in Figure 1C depicts multistage degradation. Thermogravimetric analysis of the polymer reveals initial rapid weight loss to about 10% at temperature 107.13 °C and further gradual loss of 6% between temperature ranges of 107 to 197 °C. This may be attributed to the initial degradation of polymer chains at 187 °C. It follows a steep weight loss to 14% contributing to the total 30% loss of weight. Further decline in weight started at this temperature until complete degradation at 484 °C. X-ray diffraction analysis of the polymer revealed only one peak at 28.87 °C (Figure 1D), as there exists an inverse relationship of the width of the peak to the crystal size. A thinner peak usually represents bigger crystals, but here a broader peak depicted that there might be small crystals, a defect in crystal or the amorphous nature of polymer, or it might be solid and lacking true crystallinity. Crystallinity refers to the arrangement of regular 3D structural order in a molecule or material, and polymer crystallinity increases its strength as in this phase the intermolecular bonding is significant. Moreover, increasing the degree of crystallinity increases hardness and density. Similarly, crystal size-morphology is an important property influencing the end-use applications as well as interacting strongly with the crystallization process itself. Furthermore, crystallinity and the size-morphology are not only related to solubility; however, crystallinity decreases solubility. Hydrophilic polymers are usually amorphous and have capability to enhance the solubility of many hydrophobic drugs. The SEM analysis revealed the irregular shape of the polymer′s particles (Figure 1E,F).

### 2.2. Toxicity Parameters

The biochemical and hematological parameters after administration of Ocicum are enlisted in Table 1 and RFT and LFT markers in Table 2 while the micrographs of histological examination of mice taken at magnification 14-10× after induction of 2000 mg Ocicum, as per the study protocol is given in Figure 2. At all doses, initial (5 mg/kg) and higher (50, 300 and 2000 mg/kg) all the animals survived for the study period (14 days) without signs of alteration in alertness and the skin and fur. After induction of 2 g of Ocicum, the mouse liver showed normal morphology of portal vein and hepatocytes, kidney exhibited normal morphology of renal tubules and glomerulus and the heart demonstrated normal striations with no tissue changes as expressed in Figure 2. The maximum amount of Ocicum added in solid dispersion was 300 mg, which was far lower than the maximum dose given in the toxicity study.

### 2.3. Characterization of Solid Dispersions

#### Fourier Transform Infrared Spectroscopy of Solid Dispersions

Infrared spectra of ACF showed characteristic peaks that were not only helpful indicators of drug purity but might also predict the possible drug–carrier interactions. FTIR was conducted at spectral range of 400 to 4000 cm^−1^. The ACF as well as SE-ACF-A, SE-ACF-B and SE-ACF-C in Figure 3A showed peaks at 1712 cm^−1^ due to C=O, at 1252 cm^−1^ due to C-O-C stretch and at 746 cm^−1^ due to C-Cl stretch. However, the infrared band appearing at 3320 cm^−1^ in pure aceclofenac was assigned to N-H stretches and occurred at the same value, i.e., 3315 cm^−1^ in SE-ACF-A, SE-ACF-B and SE-ACF-C. The reported band position of the N-H stretch is within 3500–3180 cm^−1^, C=O stretch at 1725 and C-Cl between 785–540 cm^−1^, as reported [24,25]. 

Poloxamer-fabricated solid dispersions of ACF prepared by the solvent-evaporation method as shown in Figure 3B displayed an N-H stretch at 3320 cm^−1^ and exhibited C=O stretches at 1717, 1712, 1717 cm^−1^ for SE-ACF-1, SE-ACF-2 and SE-ACF-3, respectively. Moreover C-O-C peaks for all three formulations occurred at 1243 cm^−1^ and C-Cl peaks were illustrated at 745 cm^−1^ for SE-ACF-1, SE-ACF-2 and SE-ACF-3.

In the melt method, as shown in Figure 3C,D, the C=O stretch at 1717 cm^−1^, the C-Cl peak at 745 cm^−1^ and N-H band at 3320 cm^−1^ remain at same position both for Ocicum and Poloxamer-constructed dispersions, whereas in lyophilized dispersions a shifting of peaks is observed in poloxamer and Ocicum-loaded dispersions as expressed in Figure 3E,F. Overall FTIR patterns of the pure drug remain similar in Ocicum-fabricated solid dispersions prepared by the solvent-evaporation method and melt method, and are somewhat similar in Poloxamer-fabricated solid dispersion prepared by the solvent-evaporation and melt methods.

### 2.4. X-ray Diffraction Analysis of Solid Dispersions

Comparative XRD analysis of ACF and solid dispersions based on preparation methods is given in Figure 4. In coherence with literature [24], characteristic peaks of drugs occurred at 8.80°, 17.49°, 22.20° and 25.94° at 2θ values with intensities of 85.02, 561.56, 67.94 and 1796. X-ray diffraction patterns of SE-ACF-B and SE-ACF-C (Figure 4A) also exhibited sharp peaks at 8.68°, 22.08°, 17.74° and 26.433° and at 8.80°, 22.32°, 17.74° and 26.30° with intensities at 97, 266, 116 and 908 and at 54.287, 110, 931, respectively, for the above-mentioned samples, whereas in SE-ACF-A, a complete amorphous pattern was observed and a remarkable reduction in peak intensities was observed in SE-ACF-B and SE-ACF-C. Likewise, solid dispersions prepared by the melting method (Figure 4E) displayed high peak intensities of ACF at 17.49° and 25.94° with 516 and 1721 intensities for M-ACF-A; at 17.61° and 26.07° with 638 and 1744 intensities for M-ACF-B; and for M-ACF-C, 17.74° and 25.94° with 649 and 1798 intensities. Similarly, lyophilized dispersions (Figure 4B) exhibited well-defined peaks at 17.49° and 25.83° for LP-ACF-B and at 17.91° and 26.01° for LP-ACF-C with considerably reduced intensities.

The X-ray diffractogram of Poloxamer included solid dispersions manifested characteristic peaks at 17.85° and 26.30° with 220 and 627 intensities for SE-ACF-1 and 17.49° and 26.30° at 161 and 363 intensities for SE-ACF-2 and at 17.64° and 25.94° at 102 and 200 intensities for SE-ACF-3 (Figure 4C). Similarly, solid dispersions of ACF prepared by melting method (Figure 4D) showed peaks at 17.74° and 26.183° with 283 and 868 intensities for M-ACF-1; at 17.49° and 26.22° with 156 and 279 intensity values for M-ACF-2; and at 17.49° and 26.18° with 84 and 293 intensities for M-ACF-3. Similarly, peaks occurred at 17.37°, 18.82° and 18.94° and 25.83°, 25.83° and 25.83° for LP-ACF-1, LP-ACF-2 and LP-ACF-3 (Figure 4E) with considerably reduced intensities. To conclude, the solvent-evaporation method resulted in remarkably reduced intensities with Ocicum and Poloxamer 407, whereas the melting method has shown prominent result with Poloxamer only; as already reported in literature, Poloxamer 407-fabricated solid dispersions by melt method had exhibited a significant reduction in intensities [17], whereas XRD patterns of Ocicum-inducted solid dispersions are exclusive to this research work.

### 2.5. Differential Scanning Calorimetry (DSC) of Solid Dispersion

A DSC study was conducted to elucidate the possible interaction of polymers with the drug. Comparative DSC analyses of ACF, Ocicum, ACF-Ocicum and ACF-Poloxamer are given in Figure 4G. The DSC curve for ACF revealed an endothermic peak at 161 °C, attributable to the melting point of ACF. In the ACF-Ocicum combination, the peak occurred at 159.51 °C, indicating the absence of any chemical reaction of ACF. A slight shift in the melting peak depression of ACF-polymer might be attributed to the partial interaction of the drug with the hydrophilic group present in Ocicum. However, the ACF-Poloxamer combination showed an absence of endotherm at 161 °C, and a peak appeared at 58.97 °C corresponding to the melting point of Poloxamer 407. DSC results further supported the FTIR findings where the FTIR spectrum of ACF with Ocicum was almost similar to the pure drug, whereas Poloxamer-loaded ACF showed the changed spectra compared to the pure drug, probably implying an alteration beyond the physical change, i.e., modification in the drug chemical structure. Chemical structure is revealed by FTIR and drug excipient interaction by DSC. DSC and FTIR demonstrated an absence of any major chemical reaction between the drug and Ocicum.

### 2.6. Microscopic Analysis of Solid Dispersions

In solvent evaporation, the Ocicum-inducted dispersions revealed spherical symmetrical clusters with individual particles with a size range of 3 to 4 µm in SE-ACF-A and SE-ACF-B, while Poloxamer-encapsulated dispersion, SE-ACF-2, showed aggregates of larger and smaller particles exhibiting polygonal shapes (Figure 5A–C). Melted dispersions prepared by Ocicum exhibited a majority of the needle-shaped particles, while others were irregular-shaped particles ranging from 10 µm to the lower ranges of 4–6 or 3–4 µm in M-ACF-B and M-ACF-C (Figure 5D,E). In the Poloxamer-fabricated solid dispersions, melted conglomerates with some individual particles having pointed polygonal irregular shapes were observable. The size of individual particles was approximately 8 to 15 µm in M-ACF-2 and M-ACF-3 (Figure 5F,G). Based on higher solubility results, these dispersions were selected for SEM illustrations.

### 2.7. Aceclofenac Content in Solid Dispersions

ACF content ranged from 95 to 110% in Ocicum and 90 to 104% in the Poloxamer-407-based solid dispersions. However, except for the Poloxamer-lyophilized dispersions, the content was within the compendial stipulated range, i.e., 90 to 110 % for all dispersions. Drug content in Ocicum dispersions is comparable to poloxamer-fabricated dispersions in the solvent-evaporation and melt methods, but in the lyophilization method Ocicum dispersions displayed significant higher values as compared to poloxamer-constructed dispersions as expressed in Figure 6A–C.

### 2.8. Solubility of Aceclofenac

Ocicum-based dispersions exhibited enhancement of ACF solubility by 2.51–3.72, 3.3–4.51 and 3.35–6.27 fold, prepared, respectively, by solvent evaporation (Figure 6D), melt method (Figure 6E) and lyophilization (Figure 6F) in water, relative to pure drug. Similarly, the Poloxamer 407-composed dispersions prepared by solvent evaporation and melt method, increased solubility 2.15–3.77 and 2.10–4.51 fold as compared to pure ACF, respectively (Figure 6D,E), yet with lesser magnitude than that of the Ocicum-based solid dispersions. The polymeric surfactant and hydrophilic carrier improve the wettability and solubilization of ACF because of the molecular structural closeness of the water-insoluble drug and hydrophilic polymer, as reported. The usual existence of the drug as an amorphous form in polymer-based dispersions also results in better solubility. The literature also supports the solubility improvement of many poorly soluble BCS class II drugs with Poloxamer 407 [17,18,19,26]. Contrarily, lyophilization could not improve solubility of ACF with Poloxamer 407 (Figure 6F), which was in line with a previous report for solid dispersion of silver sulphadiazine, fabricated with Poloxamer 407 employing lyophilization, which did not remarkably improve solubility. The highest solubility was resulted in the dispersions prepared using Poloxamer 407 in the melt method [18]. The present report documents the enhanced solubility of ACF in Ocicum fabricated solid dispersions for the first time.

### 2.9. Dissolution of Aceclofenac from Solid Dispersions

Based on higher solubility, XRD patterns and FTIR scans of SE-ACF-A, SE-ACF-B, SE-ACF-2, M-ACF-2, M-ACF-3, M-ACF-B and M-ACF-C dispersions were selected further for dissolution studies. Solid dispersions of Poloxamer 407 prepared by lyophilization manifested low drug content and solubility, while the lyophilized dispersions prepared with Ocicum demonstrated high solubility, yet exhibited changed FTIR spectra; therefore the solid dispersions prepared by lyophilization were not studied for drug release. Comparative dissolutions of ACF-Ocicum and ACF-Poloxamer solid dispersions in aqueous media and phosphate buffer are given in Figure 6G,H. Pure drug showed a 26% release in 120 min in water, consistent with the already-reported release of aceclofenac in water [27]. Ocicum-based solid dispersion prepared by solvent evaporation with drug–polymer ratio 1:1, (SE-ACF-A), 1:2 (SE-ACF-B) and melting method with ratio 1:3 (M-ACF-C) demonstrated 4-fold faster dissolution at 60 and 80 min than that of the pure drug. The T80 values of aceclofenac from Ocicum-fabricated dispersions, SE-ACF-A, SE-ACF-B, M-ACF-B, and M-ACF-C were 61.65 min, 61.99 min, 88.54 min and 79.32 min, respectively, compared to 1417 min of the pure drug. Poloxamer 407-based dispersions revealed up to 3-fold higher release when compared to the pure drug at 80 and 100 min. The values of T80 of SE-ACF-2, M-ACF-2, M-ACF-3 were 123.48 min, 125.18 min and 93.80 min, respectively. A higher release from the drug might be ascribed to the Ocicum, which improved the drug wetting due to its hydrophilic nature, which in turn was responsible for lowering the interfacial tension between the drug and dissolution medium and also prohibited the reaggregation of drug particles, thus facilitating a larger surface area. The hydrophilic polymer might also restrict the crystal growth of ACF, which contributed to the faster drug dissolution. 

In phosphate buffer, pH 7.4, SE-ACF-A, SE-ACF-B, M-ACF-B and M-ACF-C demonstrated higher dissolution profiles at all-time points especially at 10 min, 30.44%, 20.51%, 31.76% and 30.44%, respectively, and at 20 min, 35.74%, 31.43%, 22.50% and 26.80%, respectively, greater release than the pure drug. However, in the case of Poloxamer-based dispersions, SE-ACF-2, M-ACF-2 and M-ACF-3 manifested 9.26%, 16.54% and 20.51% greater release than that of the pure drug at 40 min, but lower when compared to Ocicum-formulated dispersions. 

Aceclofenac pure drug exhibited T80 of 52.62 min, and SE-ACF-A, SE-ACF-B, M-ACF-B and M-ACF-C demonstrated 16.53 min, 20.15 min, 26.43 min and 22.82 min, respectively. Likewise, SE-ACF-2, M-ACF2 and M-ACF-3 demonstrated T80 of 41.57 min, 30.90 min and 24.382 min, respectively. The release of aceclofenac was increased by increasing polymer ratios and a slightly higher drug release was observed with 1:3 dispersions when compared to the other ratios. Though the increase in dissolution of aceclofenac in the present experiments was in accordance with the earlier-reported higher dissolution of aceclofenac with Poloxamer 407 [16], yet the magnitude of the increased drug release was higher in solid dispersions prepared using Ocicum than that prepared using the Poloxamer 407. However, the improved dissolution profiles of ACF with Ocicum are exclusive to this research work.

In water, the best-fitted model for all the dispersions was Korsmeyer–Peppas, except for dispersions prepared using solvent evaporation, i.e., SE-ACF-A, SE-ACF-B and the pure drug that followed the Weibull model. Similarly, in the phosphate buffer the Korsmeyer–Peppas model best described all dispersions, except those prepared using solvent evaporation; SE-ACF-B and melting M-ACF-3 assumed first order, pure drug and dispersion with solvent evaporation SE-ACF-A, and were prepared using the melting method M-ACF-2, described by Weibull model. 

## 3. Materials and Method

Following materials were used in this study; aceclofenac (gifted from Highnoon Laboratories Limited, Lahore, Pakistan), Poloxamer 407 (Sigma Aldrich, St. Louis, MO, USA). Seeds of *Ocimum basilicum* were purchased from local market. Monobasic potassium phosphate was obtained from Duksan Pure Chemicals and NaOH from Merck Chemicals, Germany (both these chemicals were used in preparation of phosphate buffer). Formalin was purchased from Millipore Sigma. All other reagents used were of analytical grade. 

### 3.1. Study Design

The scheme given in Figure 7 was employed for performing the current study.

### 3.2. Extraction of Polymer from O. basilicum

For extraction of polymer, method described in literature was used with some modifications [28]. Briefly, the cleaned seeds of *Ocimum basilicum* were soaked in distilled water at a water–seed ratio 40:1 at pH 7 for 5 h, followed by stirring using Mophorn overhead stirrer (Model No. 001) for 15 min at 25 °C. Mucilage obtained was centrifuged at 3000 rpm for 15 to 20 min for the sedimentation of seeds with the appearance of supernatant gel containing mucilage followed by decantation. Mucilage was dried at 60 °C in conventional hot air oven until completely anhydrous. These dried layers were scrapped and powder was stored in a cool and dry place in an airtight container protecting it from light and humidity. 

### 3.3. Characterization of Ocicum

Polymer Ocicum was subjected to different analytical tests in order to characterize it. 

### 3.4. Measurement of Yield

The extracted polymer, Ocicum was weighed and % yield was calculated by the following formulae [28]:(1)$$\%\;\mathrm{Yield}=\frac{\mathrm{Weight}\;\mathrm{of}\;\mathrm{extracted}\;\mathrm{polymer}\;\mathrm{after}\;\mathrm{drying}}{\mathrm{Weight}\;\mathrm{of}\;\mathrm{basil}\;\mathrm{seeds}\;\mathrm{taken}\;\mathrm{for}\;\mathrm{extraction}}\times 100$$

### 3.5. Melting-Point Determination of Polymer

Polymer was added in capillary tube and placed in melting-point apparatus (Galvano Scientific MP-ID, Lahore, Pakistan). Melting temperature was noted as three readings to calculate the mean value. 

### 3.6. Fourier Transform Infrared Spectroscopy (FTIR)

FTIR spectra were employed to determine the chemistry of polymer and to determine the uniformity among different production batches of polymer by using FTIR spectrophotometer IR prestige-21 by Shimadzu, USA. Spectra range was between 500–4000 cm^−1^ and at 2 cm^−1^ resolution.

### 3.7. X-ray Diffraction of Powder (XRD)

To determine the crystalline physiognomies of extracted polymer, X-ray diffraction patterns were obtained using Bruker X-ray diffractometer Model D-8 (Billerica, MA, USA) discover using CU K light source which operated at 40 mA and 30 kV. All the data were obtained from 2θ, an angle of 5 to 60° at a step size of 0.02 and scanning speed of 4°/min. 

### 3.8. Thermal Analysis

Differential scanning calorimetry (DSC) and differential thermal analysis (TGA) of polymer were recorded using TA instruments DSC Q2000 and Thermogravimetric Analysis (TGA), Q500 systems (Lukens Drive, New Castle, DE, USA), respectively. During analysis heating rate of 10 °C/min under 40 mL/min purge of nitrogen was maintained.

### 3.9. Scanning Electron Microscopy (SEM)

Morphological characteristics were observed using SEM (JEOL-JSM-6480lv), Akishima, Tokyo, Japan. Carbon adhesive tapes were used for mounting of samples, which were anchored on aluminum stub. Samples were photographed after blowing with air at voltage of 2 kV for 25 sec. 

### 3.10. Toxicity Studies of Polymer

Acute toxicity studies of the extracted polymer were performed in female albino mice according to Organization for Economic Cooperation and Development guidelines 423 [29] after prior ethical approval from University of the Punjab, Lahore-Pakistan (#1075B issued on 11-04-2018). Female albino mice weighing between 200–210 g and of age 6 to 8 weeks were acclimatized to the artificial cycles of light and dark at 25 ± 1 °C for 7 days with normal supply of food (previously radiation-sterilized paddy husk) and water. The animals were randomized for control group and for administration of four doses of Ocicum, each comprising 3 animals.

Control group received only the vehicle (water, without Ocicum). The animals were fasted prior to dosing for 3 to 4 h with only water intake. Initially, 5 mg/kg of test polymer dissolved in water as vehicle was administered orally, and the mortality, if any, along with other parameters such as alertness and changes in skin and fur were assessed periodically during first 24 h, with special attention given during the first 4 h, and daily thereafter for a total of 14 days. All animals were dosed via oral gavage. If mortality was observed in two out of three animals, then the dose was considered toxic. If mortality was observed in only one out of three animals, then the same dose was to be repeated to confirm the toxic dose. As no mortality was observed after the initial dose, higher doses, i.e., 50, 300 mg/kg and 2000 mg/kg of test polymer were therefore employed for further toxicity studies. Blood samples were taken for hematological assessment on 14th day in tubes and analyzed for complete blood count (CBC), and markers for renal function test (RFT) and liver function (LFT). According to the protocol, the animals that were administered with the highest dose (2000 mg) were slaughtered at the end (on 14^th^ day and vital organs were fixed with 10% buffered formalin. Then, hematoxylin and eosin staining was carried out for histological analysis [30].

### 3.11. Fabrication of Aceclofenac-Loaded Solid Dispersions

Solid dispersions of aceclofenac were fabricated with Ocicum or Poloxamer 407 by following three methods using polymer-to-drug ratios of 1:1, 1:2 and 1:3. Thus, a total 18 formulations were prepared, 9 by each polymer with their formulation code given in Table 3. In solvent-evaporation method, the respective polymer and drug were dissolved in water as vehicle, which was then evaporated at room temperature until constant weight was obtained. In melt method, the respective polymer was heated until molten and then aceclofenac was mixed to achieve the required ratios. The molten dispersions were gradually cooled. For dispersions prepared using lyophilization, aceclofenac was dissolved in water along with the respective polymer. This solution was frozen at temperature −25 °C followed by lyophilization employing lyophilizer (Christ, Osterode am Harz, Germany), equipped with Lyo Chamber Guard until completely dry. From each method, the solid dispersions thus collected were stored in desiccator at room temperature for further characterization.

### 3.12. Evaluation of Solid Dispersion

#### 3.12.1. Aceclofenac Content Determination in Solid Dispersions

Calibration curves of ACF in phosphate buffer pH 7.4 were drawn. Phosphate buffer was prepared in accordance with method stated in USP (2008). Phosphate buffer was prepared. Solid dispersion, equivalent to 50 mg ACF, was completely dissolved in 30 mL of phosphate buffer, pH 7.4 and volume was made up to 50 mL. The appropriately diluted samples with the buffer were analyzed at 273 nm using UV-Visible spectrophotometer (Shimadzu UV2550) and the ACF content in dispersions was calculated by the following formula:(2)$${\mathrm{ACF}}_{c}=\frac{{\mathrm{ACF}}_{a}}{{\mathrm{ACF}}_{t}}\times 100$$
where ACF_c_ is aceclofenac content, ACF_a_ is actual titre of ACF measured by UV-Visible spectrophotometer and ACF_t_ is theoretical concentration. Triplicate readings were noted for each formulation.

#### 3.12.2. Solubility Test of ACF from Solid Dispersions

For solubility determination of ACF-loaded dispersions, weighed quantity of each formulation was gradually added to 3 mL distilled water in microtube and vortexed to a point until not further dissolved. Then the microtube was attached with agitator and samples were continuously shaken at 100 rpm for 5 days at a constant temperature of 25 °C. Samples were then centrifuged at 3000 rpm for 5 min. A 500 µL of supernatant was collected with micropipette and suitably diluted with water and absorbance was recorded using UV-Visible spectrophotometer (UV2550 Shimadzu) as above. Same procedure was repeated thrice [6]. Solubility (mg) was calculated from calibration curve of ACF in water.

### 3.13. X-ray Diffraction (XRD) Analysis

X-ray analysis of ACF, polymers (Ocicum and Poloxamer 407) and solid dispersions was carried out on Bruker X-ray diffractometer (Model D-8 discover), equipped with Ni filter and scintillation detector. Each measurement was recorded in the range of 10° to 60° with step size of 0.02°/sec and 2θ was taken as scanning mode. X-ray diffractometer operates with Cu, K α light source, utilizing voltage (in tube V) of 40 KVA and current (in tube I) 40 mA.

### 3.14. Differential Scanning Calorimetry (DSC)

DSC analysis of ACF and combination of ACF with natural and synthetic polymers was performed by differential scanning calorimetry (TA Instrument DSC Q2000). Samples were analyzed after placing them in aluminum pans with subsequent sealing. Temperature was gradually increased from 25 °C to 300 °C at heating rate of 5°/min.

### 3.15. Scanning Electron Microscopy (SEM)

The morphological characteristics of ACF and solid dispersions only showing higher solubility were examined using JEOL electron microscope (Model: Scar-JSM-6480LV). Samples were fixed on aluminum stubs by carbon adhesive tape, were blown with air and then photographed at voltage of 2 kV for time duration of 25 sec.

### 3.16. Fourier Transform Infrared Spectroscopy (FTIR)

FTIR spectroscopic analysis of solid dispersions and pure drug was carried out using FTIR spectrophotometer (IR prestige-21 by Shimadzu). Small quantity of each sample was taken in crucible and scanned over the range of 500 to 4000 cm^−1^ with resolution of 2 cm^−1^.

### 3.17. Dissolution Test

Dissolution test of ACF from the selected solid dispersions with higher solubility was carried out using USP paddle apparatus (Galvano Scientific Model Beta 8 L (V_3_). Solid dispersions of weight equivalent to 50 mg of ACF were tested separately in 450 mL of water and phosphate buffer (pH 7.4), half of the normal volume because of lower amount of the formulation, in line with the literature [5]. The media were maintained at temperature at 37 ± 0.5 °C. The paddle speed was set at 75 rpm and 5 ml samples were withdrawn at 20, 40, 60, 80 and 120 min for water and 10, 20, 30, 40, 50 and 60 min for phosphate buffer, due to the varied ACF solubility in both media. Samples were filtered through syringe filter (0.45 µm) and diluted appropriately with the respective dissolution media. The ACF in unknown samples was measured using the procedure, as given for drug-content determination. The drug release (%) data were fitted to various models to compute the time for 80% drug release (T80) using DDSolver [2]. The model was selected based on the highest coefficient of relationship (R^2^) and in case of close R^2^ value, the least Akaike information criterion (AIC) was used to select the model [31].

### 3.18. Statistical Analysis

The data are presented as mean ± standard deviation (SD) where applicable or unless stated otherwise. The drug content and solubility data were analyzed using factorial analysis with factors, ratios (3 levels), polymers (2 levels) and solid-dispersion methods (3 levels) using Design Expert^®^ Ver. 13 (Stat Ease Inc. Minneapolis, MN, USA) for graphical rendering to have a better understanding of the effects of above parameters.

## 4. Conclusions

Natural polymer extracted from *Ocimum basilicum* was found safe and it amorphized the drug in solid dispersions. The Ocicum-based solid dispersions by solvent evaporation, lyophilization and melting method exhibited remarkably enhanced solubility, higher than that by the Poloxamer- fabricated solid dispersions. Lyophilization showed the least solubility enhancement in the case of Poloxamer only. Higher dissolution rates were exhibited by SE-ACF-A, SE-ACF-B and SE-ACF-2 in water and SE-ACF-A, M-ACF-B and M-ACF-C in phosphate buffer, which was more than the pure drug at all time points. Ocicum showed higher solubility and dissolution than that of the Poloxamer 407. Thus, it can be inferred that Ocicum-based polymeric solid dispersions could be propitious delivery systems, which are expected for the efficacious administration of aceclofenac and other BCS class II drugs. Ocicum could be an alternative of the synthetic solubility enhancer. The industrial viability, scalability and commercialization potential of Ocicum could be assessed.

## Figures and Tables

**Figure 1 pharmaceuticals-15-00869-f001:**
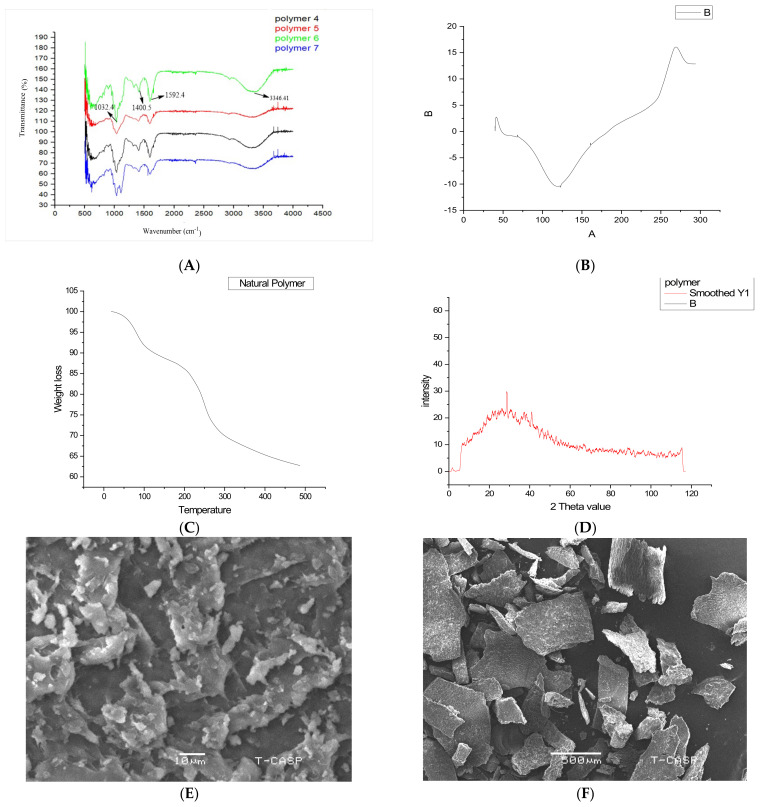
Characteristics of Ocicum: (**A**) FTIR, (**B**) DSC, (**C**) TGA, (**D**) XRD, and (**E**,**F**) SEM micrographs at different magnifications.

**Figure 2 pharmaceuticals-15-00869-f002:**
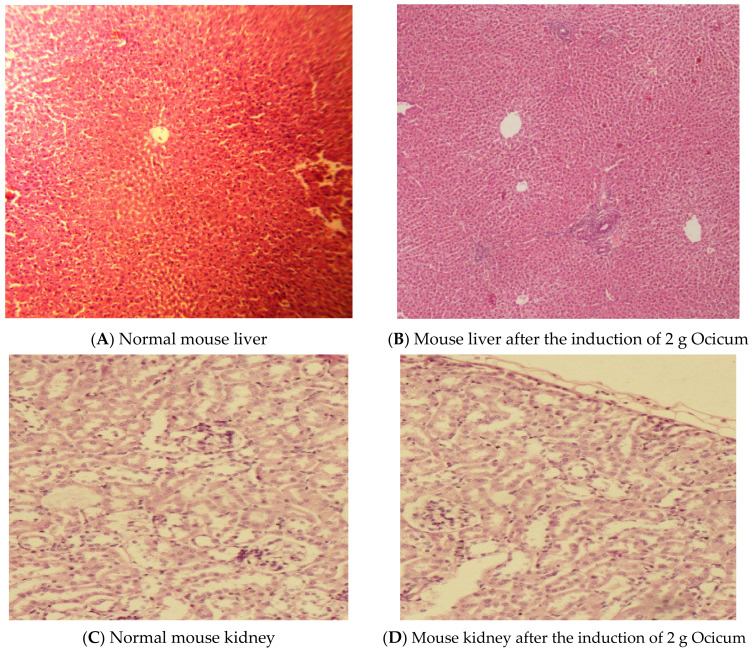
Histology of mouse liver, kidney and heart before and after the induction of 2 g Ocicum.

**Figure 3 pharmaceuticals-15-00869-f003:**
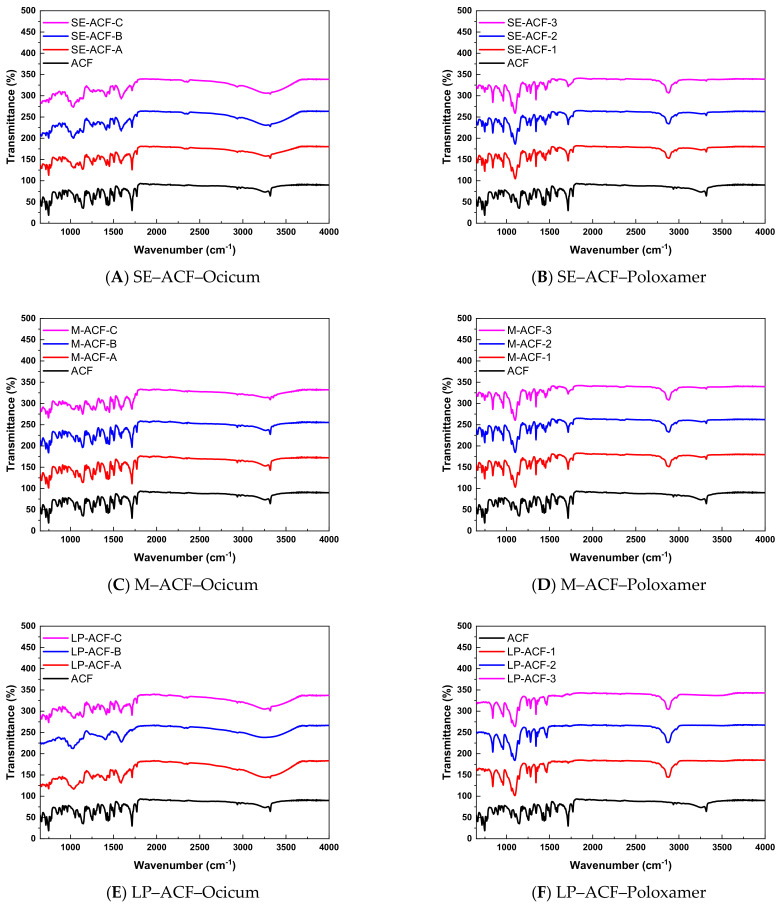
Comparative FTIR analysis of ACF and solid dispersions with Poloxamer or Ocicum in different preparation methods.

**Figure 4 pharmaceuticals-15-00869-f004:**
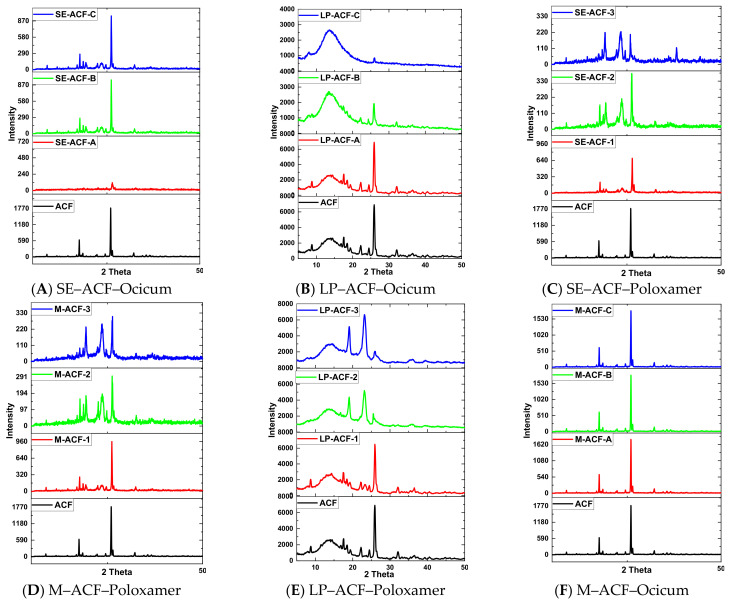
XRD and DSC of ACF and solid dispersions with polymers in solid dispersions.

**Figure 5 pharmaceuticals-15-00869-f005:**
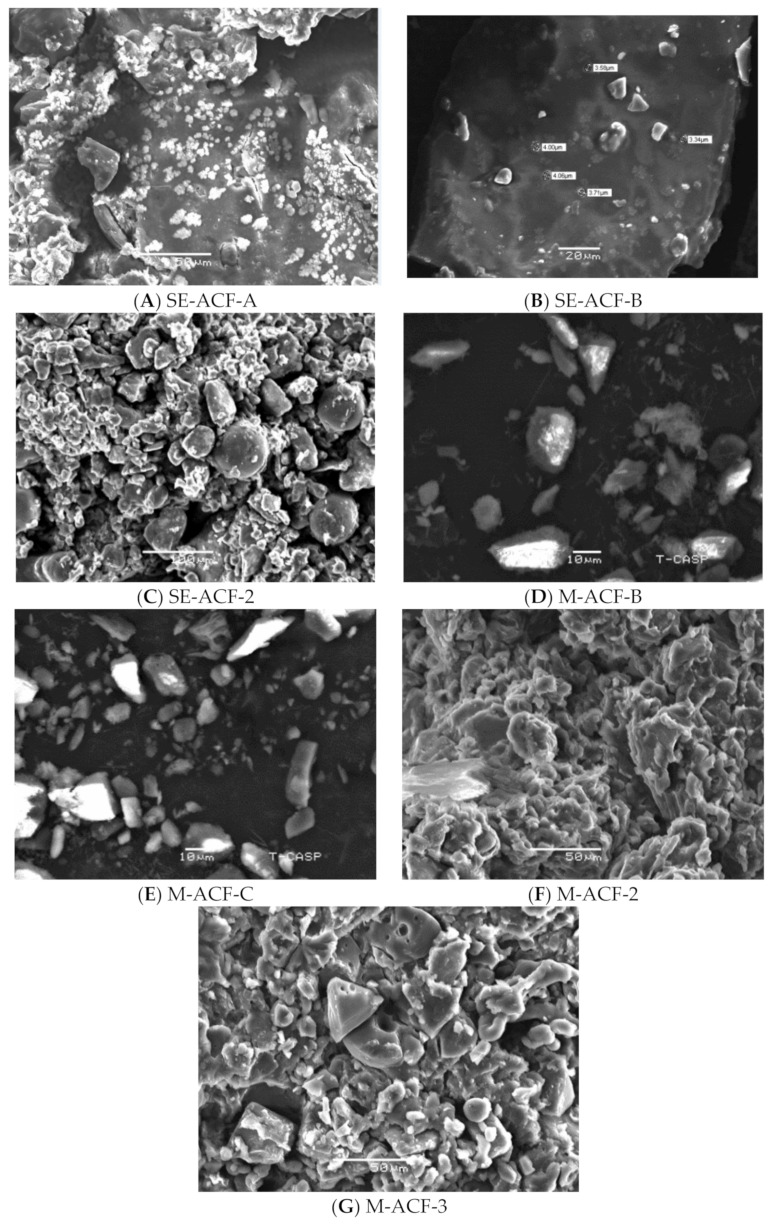
SEM of ACF solid dispersions prepared by solvent evaporation with ratios (**A**) 1:1, (**B**) 1:2 and (**C**) 1:2, and prepared by melt methods with ratios (**D**) 1:2, (**E**) 1:3, (**F**) 1:2, and (**G**) 1:3.

**Figure 6 pharmaceuticals-15-00869-f006:**
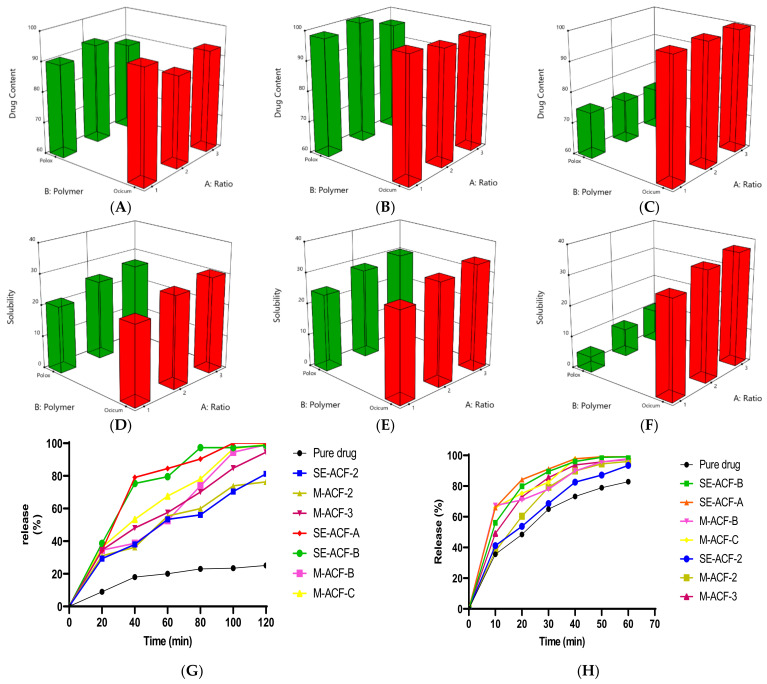
Three-dimensional plots of combined effects of polymer, their ratios and methods of preparation of solid dispersion on the aceclofenac content prepared using different methods (**A**–**C**); combined effects of polymer, their ratios and preparation methods on the aceclofenac solubility form solid dispersions prepared using different methods (**D**–**F**); dissolution of aceclofenac in Ocicum- and Poloxamer 407-based solid dispersions in (**G**) aqueous media and (**H**) phosphate buffer.

**Figure 7 pharmaceuticals-15-00869-f007:**
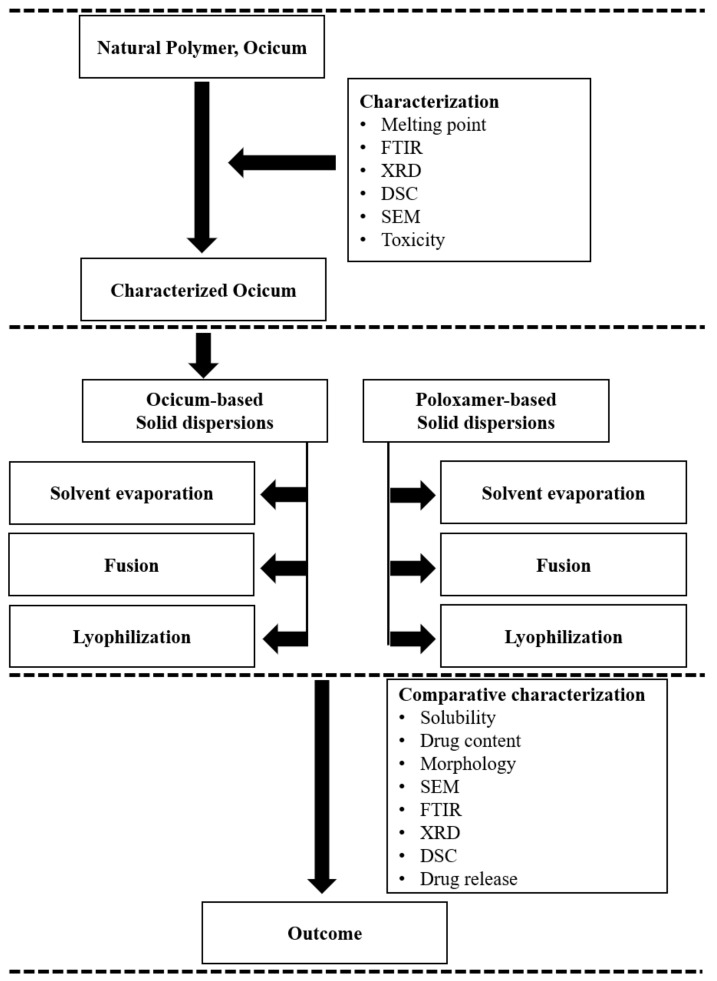
Schematics for study design.

**Table 1 pharmaceuticals-15-00869-t001:** Biochemical and hematological parameters of mouse blood after induction of Ocicum [21,22].

CBC Test Parameters	Animal Groups		Normal Scale
	Control	Polymer-Inducted	
WBC (×10^9^/L)	6.8	7	3.5–10
LYM	4.3	4	0.9–5
MID	0.8	1	0.1–1.5
GRA	1.7	3.6	1.2–8.0
MCH	18.2	22.4	25–35
MCHC	34.9	35.3	31–38
RBC (×10^6^/mm^3^)	4.54	3.98	3.5–5.5
MCV	82.4	87.9	750–1000
HCT	39.8	43.6	35–55
MPV	7.2	6.8	6.5–11
Monocyte	7	6	1–10%
Neutrophils	38	40	40–60%
Lymphocytes	32	35	18–45%

**Table 2 pharmaceuticals-15-00869-t002:** Liver and renal function parameters in control and after induction of Ocicum [23].

Test Parameters	Animal Groups		Normal Scale
	Control	Polymer-Treated	
Bilirubin total (mg/dL)	0.8	0.9	0.3–1.2
Bilirubin conjugated (mg/dL)	0.2	0.19	<0.3
SGPT (U/L)	49	53	7.0–56.0
SGOT (U/L)	28	39	5.0–40.0
Alkaline phosphatase	100	115	44.0–147.0
Creatinine	0.9	1.2	0.8–1.8
Urea (mmol)	7.3	7.5	2.1–8.5
Uric acid (mg/dL)	7.4	8.1	Adult male: 4.0–8.5
			Adult female: 2.7–7.3

**Table 3 pharmaceuticals-15-00869-t003:** Composition of solid dispersion of ACF with NP and Poloxamer 407.

Carrying Agent	Drug-to-Carrier Ratio	Formulation Code		
		Solvent Evaporation	Lyophilization	Melt Method
Ocicum	1:1	SE-ACF-A	LP-ACF-A	M-ACF-A
(Natural Polymer)	1:2	SE-ACF-B	LP-ACF-B	M-ACF-B
	1:3	SE-ACF-C	LP-ACF-C	M-ACF-C
Poloxamer 407	1:1	SE-ACF-1	LP-ACF-1	M-ACF-1
	1:2	SE-ACF-2	LP-ACF-2	M-ACF-2
	1:3	SE-ACF-3	LP-ACF-3	M-ACF-3

## Data Availability

Data is contained within the article.

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
