# Peer review of "Safety and Pharmaceutical Evaluation of a Novel Natural Polymer, Ocicum, as Solubility and Dissolution Enhancer in Solid Dispersion"

_pharmaceuticals, 2022, doi:10.3390/ph15070869_

Round 1

Reviewer 1 Report

The study explores the benefits of using a natural polymer Ocicum as solubility and dissolution rate enhancer in solid dispersions. Overall, the manuscript is well structured and scientifically sound, and can be accepted in its current form. 

Author Response

Reviewer Response Letter

The study explores the benefits of using a natural polymer Ocicum as a solubility and dissolution rate enhancer in solid dispersions. Overall, the manuscript is well structured and scientifically sound, and can be accepted in its current form.

We would like to thank the reviewer for his encouraging feedback on our manuscript.

Reviewer 2 Report

The manuscript entitled “Safety and pharmaceutical evaluation of a novel natural polymer Ocicum as solubility and dissolution enhancer in solid dispersión” presents a deep study of this polymer extracted from Ocimum basilicum in order to determine its potential as a pharmaceutical excipient. The subject of the study as well as the experimental work carried out is, in general, of interest and this is a valuable study to be published in a journal such as Pharmaceutics.

Nevertheless, in my opinion there are some issues that are not fully clear, which should be revised before considering publication:

1.- The manuscript does not indicate which is the administration route intended for this pharmaceutical. Is it oral or parenteral? Is it as a solid dispersion (for oral administration) or in solution (for either oral or parenteral)? According to the manuscript, for toxicity studies the polymer is administered in solution to the mice, but it is not indicated how it is administered. I ask for comments on this issue to be included in the manuscript.

2.- FTIR spectra in figure 2A and comments on page 8. It would help to label bands in the spectra gathered in the figure. It is explained the presence of peptidic linkages and fatty acids but it is not indicated whether the band corresponding to the carbonyl groups is present in the spectra. Should these appear in the region 1650-1580 cm-1? Neither the N-H stretching band of the amide groups. Also, peaks corresponding to stretching of C-H bonds of fatty acids should be indicated.

I deduce, but it is not clear to me, that the composition of ocicum is known and that the presence of proteins and fatty acids has been previously described, isn’t it? a brief description should be done for the easier interpretation and reading of FTIR results.

3.- page 9. Which can be the influence of the crystallinity and the size-morphology of crystals on the properties of the polymer and its application as an excipient? A comment on this should be included to understand the importance of this aspect. Is it related only to solubility?

4.- FTIR spectra in Figure 2 and 4 should better maintain the same formal presentation (wavenumber axis spears increasing in figure 2 and decreasing in figure 4)

5.- page 15. In SEM images, spherical and symmetrical clusters are not as clearly seen as claimed in the discussion. Either better images or a more accurate description should be included.

Reviewer 3 Report

The novelty of the work is to be clearly stated

more details on the extraction technique are to be added.

The variation of the melting point from 132 to 134oC is to be justified. Are the measurement effected by the surrounding conditions ?

References are to be added for the normal scales presented in tables 2 and 3.

This statement ‘’ In solvent evaporation and melt method, the drug content was higher at drug: Ocicum ratios of 1:1 and 1:3, while at all ratios in lyophillization’’ is to be explained.

How the safety is evaluated ?; the word safety exist only in the title and introduction.

Round 2

Reviewer 2 Report

Authors have addressed and answered appropriately all my comments, and I propose to publish the manuscript in its present form